# Extracellular Vesicles as Potential Therapeutic Messengers in Cancer Management

**DOI:** 10.3390/biology12050665

**Published:** 2023-04-27

**Authors:** Cristina Almeida, Ana Luísa Teixeira, Francisca Dias, Mariana Morais, Rui Medeiros

**Affiliations:** 1Molecular Oncology and Viral Pathology Group, Research Center of IPO Porto (CI-IPOP)/RISE@CI-IPOP (Health Research Network), Portuguese Oncology Institute of Porto (IPO Porto)/Porto Comprehensive Cancer Center (Porto.CCC), Rua Dr António Bernardino de Almeida, 4200-072 Porto, Portugal; 2Research Department of the Portuguese League Against Cancer Regional Nucleus of the North (LPCC-NRNorte), Estrada da Circunvalação 6657, 4200-177 Porto, Portugal; 3ICBAS School of Medicine and Biomedical Sciences, University of Porto (UP), Rua Jorge Viterbo Ferreira 228, 4050-513 Porto, Portugal; 4Fernando Pessoa Research, Innovation and Development Institute (I3ID FFP), Fernando Pessoa University (UFP), Praça 9 de Abril 349, 4249-004 Porto, Portugal; 5Faculty of Medicine, University of Porto (FMUP), Alameda Prof. Hernâni Monteiro, 4200-319 Porto, Portugal

**Keywords:** extracellular vesicles (EVs), cancer, drug delivery systems

## Abstract

**Simple Summary:**

Extracellular vesicles (EVs) are vehicles of cell communication that are able to carry several types of biomolecules derived from parent cells which, in a cancer context, can represent an oncogenic behavior, delivering them to recipient cells. Additionally, EVs present advantageous features, namely biocompatibility and high stability, which make them useful as delivery systems of new potential anticancer drugs. In fact, EVs can be used for the delivery of siRNAs, miRNAs, proteins, and nanoparticles in different diseases contexts. In this review, we highlight the potential application of EVs as drug delivery systems and their clinical applications.

**Abstract:**

A deeper understanding of the communication mechanisms of tumor cells in a tumor microenvironment can improve the development of new therapeutic solutions, leading to a more personalized approach. Recently, the field of extracellular vesicles (EVs) has drawn attention due to their key role in intercellular communication. EVs are nano-sized lipid bilayer vesicles that are secreted by all types of cells and can function as intermediators of intercellular communication with the ability to transfer different cargo (proteins, nucleic acids, sugar…) types among cells. This role of EVs is essential in a cancer context as it can affect tumor promotion and progression and contribute to the pre-metastatic niche establishment. Therefore, scientists from basic, translational, and clinical research areas are currently researching EVs with great expectations due to their potential to be used as clinical biomarkers, which are useful for disease diagnosis, prognosis, patient follow-up, or even as vehicles for drug delivery due to their natural carrier nature. The application of EVs presents numerous advantages as drug delivery vehicles, namely their capacity to overcome natural barriers, their inherent cell-targeting properties, and their stability in the circulation. In this review, we highlight the distinctive features of EVs, their application as efficient drug delivery systems, and their clinical applications.

## 1. Introduction

During cancer development, cells undergo several modifications, in which normal cells acquire pro-tumorigenic characteristics. The cell-to-cell communication network that is established is essential for cell survival and adaption to the microenvironment [1,2]. In fact, cell communication presents a key role in all biologic processes, allowing for changes in the tumor microenvironment that will be crucial for potentiating the metastasis formation at distant sites [3,4]. The communication between the tumoral cells and neighboring stromal cells begins at the earliest stages of tumor development and continues during primary growth, local invasion, intravasation, and establishment at the secondary site; it is established as a signaling network that supports tumor progression [5].

Cell communication is a process in which cells exchange signals. Traditionally, there are four main subtypes of chemical signaling in multicellular organisms: autocrine signaling, paracrine signaling, endocrine signaling, and signaling by direct contact [6]. However, in the last decades, one of the cell communication mechanisms that has gained attention, especially in the cancer field, is cell communication through extracellular vesicles (EVs) [1,2,7,8]. However, the identification of EVs as biological components with designated functions goes back 80–90 years, when several prior scientific works had already described these structures. In fact, in 1946, Chargaff and West reported the existence of EVs while studying thromboplastin and platelets, describing a specific fraction formed after spinning down blood samples with high clotting potential [9]. Later, in 1967, Wolf and colleagues also described a material fraction formed after high-speed centrifugation that originated from platelets but was different from intact platelets [9]. Subsequently, in 1971, Aaronson and co-workers used, for the first time, the term “extracellular vesicles”, showing that vesicles and other membranous structures arose from various cell organelles; the authors clearly recognized EV biogenesis as a biological process, not a fixation artefact [9]. In 1983, Pan and Johnstone were the first to discover a novel mechanism used by cells to communicate [10]. Initially, they thought that the release of EVs was part of a disposal mechanism to discard unwanted materials from cells [10]. However, successive studies have allowed us to understand that EV release is an important mediator process of intercellular communication that is present in normal physiological and pathological conditions [11,12]. In fact, EVs are shed by the cells in all three domains of life and represent the “the first form of cell cell-to-cell communication” mediated by peptides, ligands, receptors, and bioactive lipids, among others [13].

EVs consist of an umbrella term that define cell-released membrane vesicles, and it includes different subtypes [14,15]. Presently, the guidelines of the International Society for Extracellular Vesicles (ISEV) define four main subclasses, exosomes, microvesicles, oncosomes, and apoptotic bodies, depending on their biogenesis, size, and protein markers expressed in the plasma membrane (Figure 1) [16]. Exosomes are the smallest of the EVs subtypes and comprise those with sizes ranging between 30 and 100 nm [17]. This class is formed from late endosomes, which are produced by the inward budding of the limited multivesicular body (MVB) membrane [18]. Subsequently, the invagination of late endosomal membranes leads to the formation of intraluminal vesicles (ILVs) within large MVBs [19]. During the process, some biomolecules are fused into the invaginating membrane, while the cytosolic components are engulfed and enclosed within the ILVs [17,19]. Then, most of the ILVs are released into the extracellular space upon incorporation with the plasma membrane, which are then referred to as exosomes [10,20]. Microvesicles, with sizes ranging from 100 nm to 1 µm, represent a group formed by the outward budding of the producing cell plasmatic membrane [21]. Oncosomes, also referred to as large oncosomes (LOs), comprise sizes ranging between 1 and 10 µm and are a class of tumor-derived microvesicles that can spread oncogenic material to other cells and tissues in the tumor microenvironment [14,18,21]. Lastly, apoptotic bodies, with sizes ranging between 50 nm and 5000 nm in diameter, are released by dying cells into the extracellular space [21]. Concerning protein markers, exosomes express transmembrane proteins such as CD9, CD63, and CD81 and other proteins related to the plasma membrane [22,23]. The microvesicles mainly include cytosolic- and plasma membrane-associated proteins, namely the tetraspanins and proteins such as heat shock proteins, cytoskeletal proteins, and proteins containing post-translational modifications [10,20,22]. Finally, the apoptotic bodies express higher expression levels of proteins associated with the nucleus, such as heat shock protein 60 (HSP60), the endoplasmic reticulum (GRP78), histones, and the Golgi apparatus [22,24,25].

### EV Cargo

Currently, it has been accepted that EVs can transport several types of active biomolecules, transferring these molecules among surrounding cells. This cargo can include lipids, proteins, transcriptional factors, and genetic information, namely DNA, mRNA, microRNAs, and other non-coding RNAs [26,27]. Furthermore, EVs play important roles in the regulation of several physiological processes, such as cell differentiation, pregnancy, tissue development, and immune response, among others [28]. For instance, the study of Ma and co-workers showed that stem cell-derived EVs present bioactive cargo with regenerative abilities [29]. The authors mention that stem cell-derived EVs can contribute and modulate different stages of skin tissue regeneration, stimulate the inflammatory response in the wound area, and promote skin cell migration, proliferation, and (consequently) angiogenesis [29,30]. In 2017, Ferreira and colleagues reported that EVs from human adipose-derived mesenchymal stromal cells (MSCs) can activate the AKT signaling pathway, promoting dermal fibroblasts and keratinocytes migration and proliferation [31]. Additionally, as previously mentioned, EVs are also associated with pathological processes, presenting a role in pre-metastatic niche establishment and affecting tumor angiogenesis, tumor dissemination, and the induction of drug resistance [21,24,25,32]. In fact, several works have already shown that EVs derived from tumoral cells present a high metastatic potential, being able to influence the metastatic capacity of cells [22,33,34,35,36]. One pioneer study performed by Janowska-Wieczorek in 2005 showed that microvesicles derived from activated platelets induced metastasis and angiogenesis in lung cancer cells [37]. Briefly, the authors showed in A549 cells that PMVs (platelet-derived microvesicles) upregulate cyclin D2 expression, stimulate proliferation, and increase cell invasion. Moreover, PMVs stimulated mRNA expression of vascular endothelial growth factor; hepatocyte growth factor; angiogenic factors such as MMP-9, interleukin-8, and adhesion to fibrinogen; and human umbilical vein endothelial cells [37]. Moreover, Skog and co-workers found that EVs can mediate the communication between cells in glioblastoma cells. The authors observed that after exposure of the recipient cells HBMVEC (human brain microvascular endothelial cells) to glioblastoma-derived microvesicles, the Gluc mRNA levels increase in the recipient cells, having been also translated into a functional protein [38]. More recently, in 2019, Alharbi and colleagues showed that exosomes derived from ovarian cancer promote tumor metastasis in vivo, showing that exosomes derived from a cell line (exo-SKOV-3) with a high invasiveness capacity induce metastasis in vivo compared with exosomes derived from a cell line with a low invasiveness ability (exo-OVCAR-3) [39]. Thus, these studies highlight the role of the capacity of EVs to transport bioactive molecules and their effect on the cellular phenotype modification of recipient cells, modifying the cell phenotype for aggressive malignant characteristics [32,38]. Additionally, in 2018, Lin and co-workers described that in hepatocarcinoma cells, EVs of malignant tumors cells containing miR-210 can promote the tubular-like structure formation of endothelial cells, resulting in pro-angiogenic processes and increasing the tumor growth rate [40,41]. Moreover, abundant miR-210 can be packed into EVs and transferred to endothelial cells, and after being taken up by HUVECs (human umbilical vein endothelial cells), miR-210 promotes angiogenesis via the downregulation of the expression of SMAD and STAT expression [40]. Another study of Yang and colleagues found that hepatoma-derived exosomal miR92a-3p has a critical role in the EMT progression and promotes metastasis by activating AKT/Snail signaling and inhibiting PTEN [42]. In accordance with the studies described, we also found that *LAT1* mRNA is present in EVs derived from a colorectal cancer cell line, and the administration of these EVs to recipient cells is associated with changes in the cell phenotype and in the transcriptional and protein profiles of the recipient cells [33].

Therefore, a whole analysis of the cargo of EVs could be a useful strategy for clarifying their release mechanisms, activation states, and effects in different pathological conditions, allowing for their future application as molecular biomarkers [43]. Databases such as ExoCarta or Vesiclepedia are important tools that are adding information regarding the proteins, RNAs, lipids, and other metabolites identified in EVs [43,44].

It is recognized that the cargo of EVs comprises endosomal proteins and components; however, EVs also include material from other cellular compartments, namely from the mitochondria [45]. Interestingly, EVs can also carry mitochondria and mitochondrial DNA which, under hypoxia, is able to induce an inflammatory response in recipient cells [45]. Moreover, there seems to be a synergic effect between the mitochondria and EVs because, on the one hand, the mitochondria can modulate the production and release of EVs; on the other hand, however, EVs can also regulate the mitochondria function of the recipient cells. For example, exosomes derived from cancer-related fibroblasts supply diverse metabolites for the TCA cycle of cancer cells [46].

The capacity to study and understand the molecular traits of a patient’s cancer will allow us to enter a new era of precision medicine in oncology. Thus, tumor-derived EVs (TD-EVs) could be a promising tool that provides cancer information that could be applied in the liquid biopsies field; thus, they present great clinical value for application in cancer diagnosis, prognosis, and treatment response assessment.

In fact, EVs present several advantages compared with other traditional biomarkers, as they can be virtually released by practically all cell types and can be isolated and purified from all body fluids [1]. Furthermore, they can function as a mirror of the disease and affect several aspects of cancer treatment efficacy. Additionally, EVs also have the capacity to protect natural cargo from freeze/thaw cycles during long-term storage and allow for a more effective protection of disease biomarkers from body fluids [25,47,48]. One example of this is the study of Ogata-Kawata and co-workers, who identified 16 miRNAs that are significantly highly expressed in serum exosomes from CRC patients compared with healthy individuals as well as in colon cancer cell lines [49]. From those, eight miRNAs (let-7a, miR-1224-5p, miR-1229, miR-1246, miR-150, miR-21, miR-223 and miR-23a) presented decreased levels after surgical resection of the primary tumor, which highlights that the source of these exosome-derived miRNAs is the primary tumor [49]. In another study, Li and co-workers showed that the high levels of exosomal miR-21 secreted by hypoxic oral squamous cell carcinoma (OSCC) cells downregulated E-cadherin [50].

The establishment of new diagnostic and prognostic approaches using patient-derived vesicles from different body fluid types can increase the knowledge concerning the EVs content derived from tumor cells and open new possibilities for patient follow-up and the identification of new therapeutic targets [8]. An example of this is the study of Wang and co-workers, which describes that the exosomal miR-125-3p was upregulated in CRC patients compared with healthy individuals [51]. Additionally, González and colleagues showed that high levels of exosomal miR-19a in CRC patients was associated with lymph nodes, liver metastasis, and tumor infiltration [52]. Moreover, the study of Chiba and co-workers demonstrates that the EVs secreted from human CRC cells can transfer RNAs into liver cells, affecting the regulation of gene expression, cell intravasation, invasion, and metastasis capacity [34]. Moreover, Dias and co-workers analyzed the dynamic of a panel of nine EV-derived miRNAs in clear cell renal cell carcinoma (ccRCC) patients during the course of the disease [53]. The authors observed that the levels of EV-derived hsa-miR-25-3p, hsa-miR-126-5p, hsa-miR-200c-3p, and hsa-miR-301a-3p decreased after surgery, while hsa-miR-1293 EV-derived levels increased [53]. Additionally, the authors also found that metastatic patients present higher levels of EV-derived hsa-miR-301a-3p and lower levels of EV-derived hsa-miR-1293 than patients with a localized disease [53].

Taking this data into account, the use of EVs and their cargo analysis could improve the discovery of new biomarkers that can be applied in a liquid biopsy approach, providing helpful information for cancer diagnosis, prognosis, and the follow-up of treatment and allowing for a personalized approach.

## 2. Therapeutic Enriched Drug EVs

The rising burden of cancer as a global health issue has led to an the research field investigating the discovery of more effective anticancer therapeutic molecules and new therapeutic delivery systems [54,55,56]. There are several parameters that significantly limit the use of drugs in clinical practice. For example, several compounds present low uptake efficiency, low therapeutic index, low solubility, hepatic disposition, or unfavorable pharmacokinetic parameters [57,58]. Thus, synthetic delivery systems have been developed; unfortunately, these seem to cause diverse and severe side effects, including liver toxicity [59,60,61,62]. On the other hand, EV-based delivery systems present less cytotoxicity and are more efficient compared with other synthetic methods as they present the ability to cross blood barriers, presenting high biocompatibility while exhibiting low immunogenicity [63].

In the last few years, the biomedical field has been expanding due to the promising developments of nanotechnology being applied to medicine. The application of nanotechnology in cancer treatment opens the opportunity for the discovery of new target therapies that directly and selectively target tumor cells. Specifically, using these delivery approaches, it will be possible to conduct a more precise targeting of cancer cells in a safer and more efficient manner [64,65]. Nanoparticles offer a promising methodology for prolonging the circulating time rate and for efficient drug biodistribution [66]. However, some of these approaches present limitations, including the clearance by the immune system and impaired diffusion in the tissue microenvironmental [67]. In fact, the literature reports nano-antioxidant-based methods as a promising strategy for disease treatment applications [68]. Still, their low absorption, degradation during drug delivery, and difficulty to cross cell membranes are some of the limitations identified [69,70]. Additionally, studies highlight that some toxicity develops, as well as the formation of some nanocomposites and dangerous metabolites, which are issues that must be surpassed in order to promote patient health in a global and holistic perspective [71]. Recently, Ahmandian and coworkers showed that the capacity of antioxidant compounds could be improved using nanostructured lipid carrier delivery systems [72]. In fact, the authors found that these structures allow for high drug encapsulation and can prevent the toxicity induced by the herbicide paraquat [72]. A study performed by Chodari and coworkers highlights the need of novel delivery technologies in order to overcome the weak pharmacokinetics properties of polyphenol compounds and their delivery to target tissues [73]. Thus, the development of new delivery strategies is imperative for improving tissue-specific delivery in a safe and effective manner.

Recently, endogenous nanoscale membrane vesicles have received attention because they present characteristics such as a micro/nanostructure and bioactive composition, which could answer some of the limitations mentioned for the traditional delivery systems [74]. Furthermore, EVs have been presented to contain an intelligent core, allowing them to respond to endogenous and/or exogenous signals, target sites of disease, and provide treatment feedback for best function in patients [74,75]. These developments represent a new way for next-generation nanomedicine and offer new opportunities for drug delivery systems.

### 2.1. Effects of EV Cargo Components in Target Cells

EVs are cell-secreted nanoparticles that are recognized as natural carriers for the short- and long-distance transportation of molecules and that have an innate biocompatibility, non-cytotoxicity, and capacity to target specific cells [64]. Furthermore, because of their intrinsic tissue-homing capabilities, they are being explored for the delivery of therapeutic cargo to specific cells or tissues [76]. In fact, their characteristics have introduced them as naturally occurring and promising drug delivery systems for cancer treatment [64,77]. Additionally, EVs present a CD47 marker, which could work as a “do not eat me” signal and offer protection against the immune system [78]. Indeed, CD47-enriched EVs were demonstrated to escape phagocytosis and presented an increase in RNA accumulation into tumor cells [78,79]. A study of Kamerkar and colleagues reports that siRNA or shRNA drugs specific to oncogenic KrasG12D mutation were electroporated into human fibroblast CD47-enriched EVs and led to an inhibition of tumor proliferation in different in vivo models of pancreatic cancer, with a significant improvement in the survival rate [78,80,81]. Additionally, studies have exploited the potential use of EVs for drug delivery/targeting for theranostic usage after being engineered for drug loading [82,83,84,85,86]. In fact, different drugs, namely gemcitabine, doxorubicin (DOX), or paclitaxel (PTX), can be encapsulated in EVs, and their administration in pancreatic, colorectal, and prostate cancer cells showed antitumor activity [87,88,89,90,91]. In 2014, Pascucci and co-workers were pioneers, showing that MSCs are able to package and deliver active drugs through their MVs, which suggests that MSCs could function as a factory and can be applied for drug delivery with a higher cell-target specificity [92]. Moreover, Hadla and colleagues in 2016 found that exosomes increase the therapeutic index of DOX in two mouse cancer models, namely breast and ovarian cancers [93]. The authors demonstrated that exosomal-loaded DOX is safer and more effective than free DOX, and this delivery system was associated with an increase in DOX efficacy in both the breast cancer mouse model and in the high-grade serous ovarian cancer immunocompetent mouse [93].

Recently, González-Sarrías and co-workers reported that milk-derived exosomes can act as nanocarriers to deliver curcumin to breast tissue and, consequently, enhance their anticancer activity [94]. Additionally, Besse and co-workers studied dendritic cells-derived exosomes (Dex) as immunotherapy modulators after first-line chemotherapy in non-small cell lung cancer (NSCLC) [95]. The authors investigated if the second generation Dex (IFN-γ-Dex) could function as NK and T cell immune response boosting and confirmed that Dex presents the ability to boost the NK cell arm of antitumor immunity in patients with advanced NSCLC [95].

On the other hand, much evidence has shown that the delivery of biomolecules, such as RNAs or proteins, through EVs can also be used as a tumor therapy [96]. In fact, Kobayashi and co-workers demonstrated that ovarian cancer cell-derived EVs loaded with miR-199a-3p led to the downregulation of its target gene, the mesenchymal-epithelial transforming factor (c-Met) mRNA, and the EVs caused the inhibition of tumor cell proliferation and invasion ca-pacity [97]. Moreover, a study by Liang and colleagues showed that engineered EV-based fluorouracil (5-FU) and a miR-21 inhibitor oligonucleotides delivery system could efficiently facilitate the cellular uptake and significantly downregulate miR-21 expression in 5-FU resistance HCT 116 cell lines [82]. The authors found that these engineered EVs had the ability to revert cancer cell resistance to the 5-FU and enhance antitumor cytotoxicity [82]. Another study by Katakowski and co-workers showed that MSCs transfected with miR-146b expression plasmid released exosomes that, when injected in tumors, reduced the glioma xenograft growth in a rat model of a primary brain tumor [98]. Wahlgreen and colleagues developed a system using human exosomes to deliver siRNA into T cells and monocytes [99]. In their study, exosomes were isolated from diverse cell types, including HeLa cells, lung cancer cells, and TB-177 cells. Subsequently, siRNA was loaded into exosomes via chemical and physical transfection methods. According to the results, the siRNAs were successfully incorporated in exosomes, and their delivery induced a posttranscriptional gene silencing in the receptor cells [99]. Moreover, the authors demonstrate the successful gene silencing through the downregulation of the specific gene, as exosome-loaded siRNA leads a decrease in tagged siRNA against mitogen-activated protein kinase 1 (MAPK-1) expression [99]. Thus, these studies emphasize the potential of the application of RNA-loaded EVs in the management of the molecular mechanisms of diseases. 

### 2.2. EV-Loading Methods

As previously mentioned, therapeutic agents, including siRNAs, miRNAs, mRNAs, proteins, and chemical drugs, can be loaded into EVs (Figure 2) [15,100,101,102].

Over the years, EVs have been specifically engineered to increase their tumor-targeting capacity and drug delivery efficiency [75]. An example of such was the development of an EV-based delivery system that could target cancer stem cells with improved efficiency and specificity [103]. Considering their potential applications for human diseases, the methods for selecting and isolating EVs from parental cells as well as the cargo-loading EV approach have been the subject of several developments. The main advantages and disadvantages of the different methods are described in Table 1.

The packaging of therapeutic cargo into EVs can be performed using passive loading or active loading [107]. Currently, there are two different methods used for encapsulating cargo into EVs: Cell-based loading methods and non-cell-based loading methods [107]. In the first approach, cargo is usually loaded into isolated EVs through diffusion [115]. In the second approach, the non-cell-based loading approach refers to the direct loading of biomolecules (mRNAs, miRNAs, drugs, and proteins) into isolated EVs through sonication, electroporation, freeze/thaw cycles, incubation, extrusion, and saponin (Figure 3). These mechanisms temporarily disrupt the EVs membrane; however, this is restored after the therapeutic agent/drug is loaded [116,117,118]. In the sonication method, an ultrasound probe with different amplitudes is used to permeabilize the EVs membrane and promote drug loading [119]. This method is especially used with hydrophobic drugs and can prevent protease destruction. Lamichhane and co-workers explored the potential of sonication as a method for loading small RNAs in EVs in breast cancer cells [120]. The authors showed that MCF-7-EVs loaded with siRNA were uptaken by MCF-7 recipient cells and were able to knockdown the target mRNA, leading to a reduced protein expression [120]. Moreover, in order to develop a new exosomal-based delivery system to treat Parkinson’s disease, Haney and colleagues loaded catalase into EVs ex vivo, applying a combined treatment of 18 h of incubating saponin at room temperature (0.2%, 20 min) followed by freeze/thaw cycles (three times, 30 min), sonication (500 V, 2 kHz, 20% power, six cycles by 4 sec pulse/2 sec pause), and extrusion (the mixture was extruded 10 times) [85]. Interestingly, using just sonication, extrusion, and permeabilization with saponin resulted in a high loading efficiency, sustained release, and catalase preservation against proteases degradation [85].

On the other hand, electroporation uses an electric field to disrupt the membrane of EVs and produce temporary pores for the drugs to penetrate into EVs [119]. This method presents the disadvantage of destroying the integrity of EV membranes, which leads to a decrease in the loading efficiency. Additionally, the high voltage pulse can lead to EV aggregation [119]. In 2011, the study of Alvarez-Erviti and colleagues showed that EVs derived from dendritic cells (6–12 μg) can be loaded with exogenous siRNA (150 μg) using electroporation (400 V, 125 μF) [121]. The intravenously injected RVG-targeted exosomes delivered GAPDH siRNA specifically to microglia, neurons, and oligodendrocytes in the brain, resulting in a specific gene knockdown. Additionally, the authors also demonstrated in wild-type mice that the therapeutic potential of exosome-mediated siRNA delivery was the strong mRNA (60%) and protein (62%) knockdown of *BACE1* mRNA, a therapeutic target in Alzheimer’s disease [121].

Regarding freeze/thaw cycles, the principle of this method is mixing the EVs with drugs and freezing them at −80 °C or with liquid nitrogen, with a subsequent thawing at room temperature [119]. This method presents the disadvantage of a lower drug loading capacity. On the other hand, the incubation method has in principle a direct co-incubation between EVs and the therapeutic agent, and the loading efficiency depends on the concentration gradient of the therapeutic agent in the solution and its hydrophobicity [119]. This method is easy to use and it does not affect the integrity of the EV membrane, but it presents a lower loading efficiency. In 2010, Sun and co-workers used an incubation method and showed that the anti-inflammatory activity of curcumin is improved when encapsulated in exosomes [122]. The authors prepared exosomal curcumin by mixing curcumin with EL-4 (mouse lymphoma cell line) exosomes in PBS (150 min) [122]. The authors concluded that mice treated with curcumin (20 μmol/l) complexed with exosomes are protected against lipopolysaccharide (LPS)-induced septic shock and that these exosomes are taken up by activated monocyte-derived myeloid cells circulating in the peripheral blood, which induces the apoptosis of these monocytes [122].

In the extrusion process, it is necessary to use an extruder to mix the EVs with therapeutic agents. This method is associated with a high drug loading efficiency, but it causes damage to the plasma membrane structure of EVs [119]. Saponin is a permeabilizing agent that is capable of forming a complex with cholesterol present in the cytoplasmatic membrane of EVs in order to form a porous structure in the surface of the membrane, stimulating the incorporation of drugs [119]. To implement this method, the concentration of saponin used should be minimal and EVs must be washed (with PBS for example) after being incubated [123].

One of the main disadvantages of nanomedicine is the toxicity that drugs can cause. However, when we talk about toxicity using EVs, they seem to be rare. In fact, Tofolli and co-workers have tested the toxicity of exosomal doxorubicin (15 mg/kg) compared with free doxorubicin, concluding that exosomal doxorubicin loaded using electroporation presents less toxicity in both in vitro and in vivo doxorubicin through an altered biodistribution; this conclusion was made as the heart of mice presented a 40% reduction in exosomal doxorubicin while still presenting a similar antitumor effect [124]. Moreover, Schindler and co-workers observed that exosomal delivery of doxorubicin allows for a rapid cell entry and enhanced in vitro cytotoxicity, and it is able to prevent cardiac side effects [125]. Recently, Fan and colleagues proposed a decoy exosome system based on mesenchymal stem cell exosomes with a DNA nanostructure that did not affect the doxorubicin effect and reduced the chemotherapy-induced toxicity [126].

### 2.3. Autologous EVs

Recently, the use of autologous exosomes for drug delivery is one of the most interesting applications of these structures in cancer treatment [127]. EVs originating from autologous cancer cells are associated with minimal toxicity, reach parental cancer cells through endocytosis, and are associated with lower immunogenicity compared with other delivery vehicles [84]. Therefore, another advantage is the potential tropism of autologous EVs to the tumor microenvironment, which makes them competitive delivery vehicles [87]. In the last few years, the use of EVs derived and isolated from cancer patients’ plasma has demonstrated a high degree of specificity to tumor tissue. Additionally, it has been proposed that these isolated EVs can be loaded with different molecules and re-administrated in the same patient [20]. One of the main advantages of using autologous EVs is that they are more “biocompatible”, and they are expected to have a low risk of immunogenicity and increase the targeting capacity, allowing for more efficient therapeutic agent delivery [76,127,128]. In fact, the study of Yong-Jiang and colleagues evaluated the targeting efficacy and anticancer effects of autologous exosomes for the targeted delivery of GEM for the treatment of pancreatic cancer [87]. The authors demonstrated that GEM loaded in autologous exosomes (ExoGEM) presented superior therapeutic efficacy against pancreatic cancer, with minimal damage to normal tissues and with prolonged survival in a dose-response manner [87]. Additionally, Villa and co-workers isolated EVs derived from patients diagnosed with cancer and used them in an autologous transplantation protocol aimed at delivering theranostic cargo to neoplastic tissues [127]. The authors showed that EVs isolated from CRC patients’ plasma were capable of recognizing tumor cells and delivering a diagnostic fluorescent agent into the neoplastic tissue when administrated by injection into patient xenograft mouse models [127]. Moreover, Dai and colleagues developed a phase I clinical trial where they used a combination of ascites-derived exosomes (Aex) and granulocyte–macrophage colony-stimulating factor (GM-CSF) in the immunotherapy of CRC, showing that immunotherapy with Aex in combination with GM-CSF was safe and could be an option for advanced CRC patients [128].

## 3. Translation to Clinic

Nowadays, EVs have several clinical applications, including their use as biomarkers, therapeutic agents, drug delivery shuttles and cancer vaccines, and they have been applied from EVs of human or plant origin, which highlights their potential in different clinic settings (Table 2).

Considering the source of the EVs, several clinical trials have applied autologous dendritic cells-derived exosomes loaded with tumor antigens in cancer patients. In fact, dendritic cells-derived exosomes present components that could function as antigen-presenting structures and are capable of promoting a cell immune response [130]. Dendritic cell-derived exosomes represent a good option for cancer vaccination, as these structures present higher stability during long periods than dendritic cells because of their lipid structure and because they are also more resistant to immunosuppressive mechanisms in the tumor microenvironment [132].

It has been described in some clinical trials that dendritic cells-derived exosomes can be manufactured from cultured peripheral blood mononuclear cells obtained from patients [133]. Briefly, according Lamparski and co-workers, exosomes can be recovered from the supernatant of monocyte-derived dendritic cells that are derived from CD14+ monocytes [133]. In fact, some clinical trials have applied this exosome production procedure [129,130]. According to Morse and co-workers, the administration of autologous dendritic cell-derived exosomes loaded with the MAGE tumor antigens was feasible and tolerated by patients with advanced non-small cell lung cancer [130]. The MAGE antigens are frequently expressed in several malignancies, and their expression is linked to pro-tumorigenic characteristics and the patient prognosis [134]. Thus, this has led to several tumor antigen-specific strategies for cancer treatment. On the other hand, Escudier and co-workers also used exosomes derived from autologous monocyte-derived dendritic cell cultures loaded with MAGE 3 peptides for the immunization of advanced melanoma patients, showing the feasibility of large-scale exosome production and the safety of these autologous exosome administration in patients [129]. However, some authors have indicated that these vaccines were associated with a limited immune response, which can be explained by the type of immature dendritic cells used; this issue can be overcome using exosomes derived from mature dendritic cells [132,135].

The results of these clinical trials led to another study (NCT01159288) that proposed an immunotherapy strategy involving metronomic cyclophosphamide followed by tumor antigen-loaded, dendritic cell-derived exosomes vaccination in non-small cell lung cancer patients; this method was proposed as metronomic cyclophosphamide is able to inhibit Treg function, restoring T and NK cell functions, and exosomes are capable of activate the innate and adaptive response. In the mentioned study, the dendritic cell-derived exosomes were upgraded in order to enhance the induced T cell responses. In fact, the exosome boosts antitumor immunity in advanced lung cancer patients in a safe manner [95]. Moreover, Narita and co-workers showed that the use of exosomes derived from dendritic cells pulsed with SART1 were well tolerated, inducing antigen-specific cytotoxic T lymphocytes [131].

On the other hand, Dai and co-workers reported that the weekly immunizations of exosomes from malignant ascites in combination with the granulocyte-macrophage colony stimulating factor in colorectal cancer patients were well tolerated and safe, inducing a beneficial tumor-specific antitumor cytotoxic T lymphocyte response [128].

Another approach of an EV-therapeutic strategy was the application of gliobastoma cells that were resected and treated with small antisense oligodeoxynucleotide and directed against insulin-like growth factor 1 receptor in patients’ abdomens (NCT01550523). The hypothesis is that tumor cells, while dying, release EVs that can trigger the activation of T cell-mediated antitumor immune response [136].

Nevertheless, considering the role of EVs as natural containers with the ability to perform the cell-to-cell transport of several active molecules, there is a special interest in engineering these structures in order to deliver key therapeutic molecules. The phase I study NCT03608631 intends to test the best dose and side effects of mesenchymal stromal cells-derived exosomes with KrasG12D siRNA in metastatic pancreatic cancer patients with KrasG12D mutation.

On the other hand, alternative exosome sources are also being explored. One example is the use of natural grape exosomes to prevent oral mucosistis development in patients with head and neck cancer that have been submitted to chemo- and radiotherapy in order to evaluate the cytokine production and immune modulation induced by these nanostructures (NCT01668849). Moreover, despite the potential benefits in immune modulation by plant exosomes, it could also be interesting to use plant exosomes as delivery systems in order to increase the bioavailability. In fact, a clinical trial will attempt to overcome the curcumin limited bioavailability observed in previous trials using plant exosomes to deliver the drug to colon tumors and normal colon tissue (NCT01294072). However, although the EVs from plants present the advantages of animal-free issues, absorption into circulation, and beneficial therapeutic effects, the information is less extensive in terms of the dynamic specific markers, and the reports/data are not complete compared with evidence from human EVs.

Nevertheless, it is important to keep in mind that exosome application in clinical settings needs to comply with specific good manufacturing practice methods, and several concerns need to be considered: the type of cells, in vitro culture environment and systems, culture media, purification procedures, and characterization and identification methods. Furthermore, it is crucial to follow the scientific recommendation of the advanced therapy medicinal products [137].

## 4. Conclusions

EVs have been recognized as key components of inter-cellular communication, having the capability to deliver different biomolecules and induce molecular and phenotypic alterations in recipient cells. Moreover, it is accepted that EVs can transfer signals over long distances to modulate several physiological and pathophysiological mechanisms.

The potential application of EVs as delivery systems has been receiving a lot of attention in the last few years. Firstly, EVs can be functionalized and personalized through the loading of different biomolecules, including proteins, lipids, nucleic acids, and drugs [138].

Additionally, EV delivery systems also present advantages compared with others synthetic delivery systems, including low immunogenicity, excellent biocompatibility, and biostability, presenting a long-term accumulation in organs or tissues and specific tropism for some cell types. Moreover, they show an extraordinary ability to interact with and accumulate in target cells, and unlike other delivery systems, they are able to overcome several biological barriers such as the cytoplasmic membrane and BBB, which makes them ideal as a therapeutic delivery approach [107]. Nevertheless, different studies have reported some issues, such as the lack of standardization of both the technology for EV production and quality control, which depend on cost-effective, large-scale production and the widely applicable methods for drug loading [15,76].

Moreover, even though different loading methods have been the subject of development in the last few years, their loading efficiency is variable and still quite low. The optimization of these methods with respect to this parameter may help to improve the efficacy of future therapeutic approaches using EVs. Additionally, although not discussed in this study, studies have already reported a major struggle regarding the large-scale production of EVs [139,140,141]. Currently, there is no agreement on the best technology for EV production or EV isolation, but these are key points that may condition the future application of EVs in the clinic [15,142,143].

Indeed, functionalized EVs have shown great potential in the biomedical field, namely in the treatment of various diseases such as cellular regeneration processes, cancer, or even inflammation associated morbidities, and many of them have entered the clinical trials phase [91,96]. In fact, recent studies have highlighted the role of MSCs for EVs production compared with other cells as they present lower immunogenicity, preventing the activation of the immune response. Additionally, the application of autologous EVs has been gaining attention in the development of new treatment approaches in addition to the application of EVs as natural containers to transport active molecules in order to deliver key therapeutic molecules. However, despite the EV-based therapies being considered feasible, their application in the clinical setting could be challenging, opening new perspectives for improving the management of clinical cancer patients in the future.

## Figures and Tables

**Figure 1 biology-12-00665-f001:**
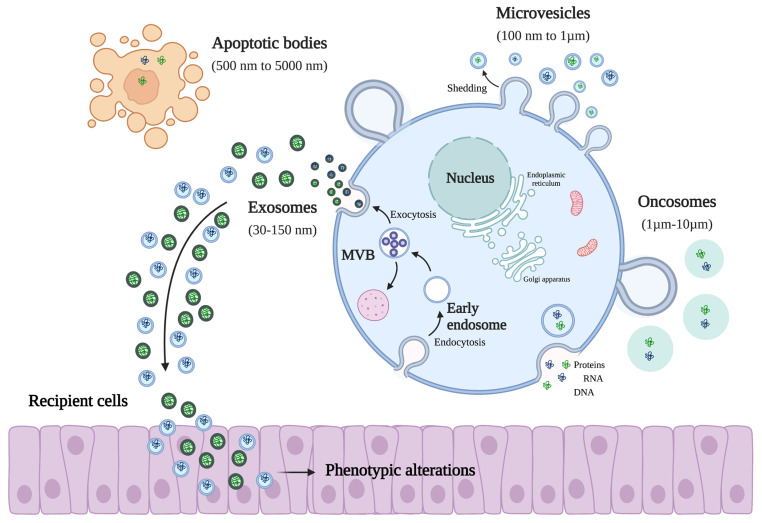
Representation of different EV subtypes and the effect of EVs in recipient cells. Extracellular vesicles (EVs) are classified into four different subclasses, taking into account their size, biogenesis, and mode of secretion. When targeting recipient cells, EVs have the ability to induce phenotypic alterations; for example, they can increase the capacity of cell proliferation or their migration ability (created with Bio-Render).

**Figure 2 biology-12-00665-f002:**
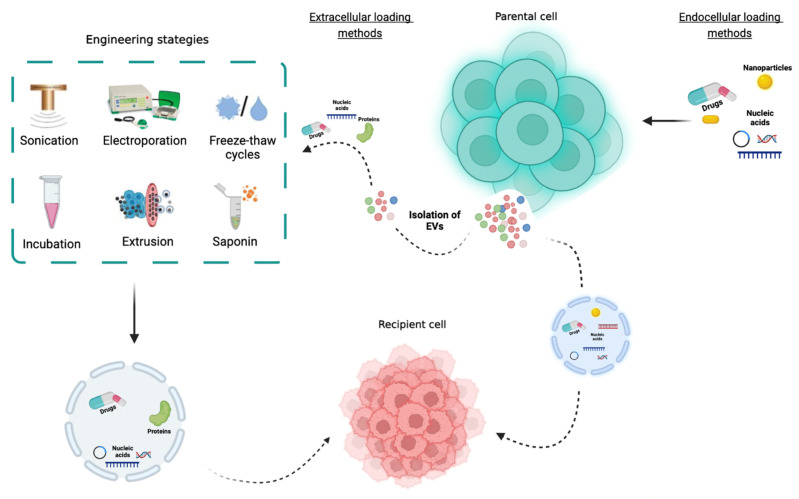
Engineered methods for loading therapeutic agents into EVs. Endogenous cargo incorporation methods modify the EV-producing cells to incorporate therapeutic agents into the EVs through the natural EV biogenesis pathways. On the other hand, exogenous cargo can be loaded into EVs using loading methods such as sonication, electroporation, freeze/thaw cycles, co-incubation, extrusion, or saponin (created with Bio-Render).

**Figure 3 biology-12-00665-f003:**
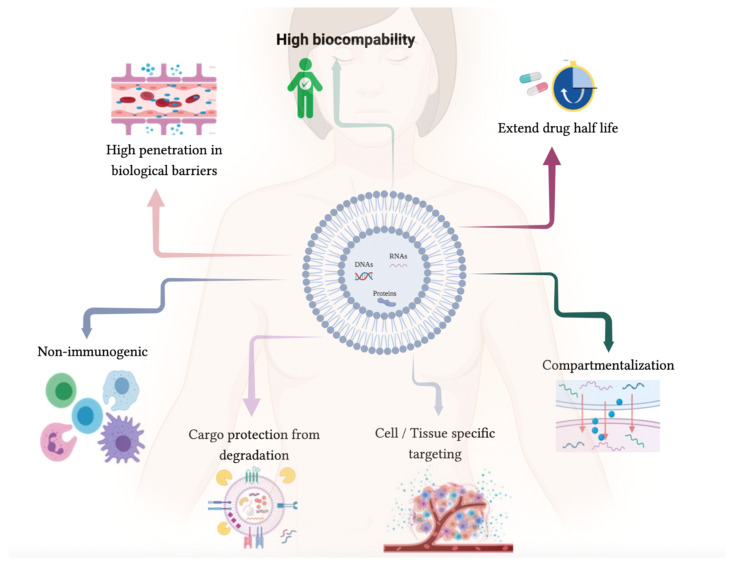
Characteristics/advantages of EVs as good candidates for being natural nano-carriers in therapeutic delivery (created with Bio-Render).

**Table 1 biology-12-00665-t001:** EVs-cargo engineered methods.

Cargo-Engineered Method	Principle	Advantages	Disadvantages	Encapsulation Efficacy (%)	Reference
**Endocellular loading methods**					
**Incubation**	Direct administration of therapeutic agents in cells	Used for small-molecule chemical drugs with low cytotoxicity	The rate of encapsulation is low	Approximately 15%	[75,104,105,106]
**Transfection**	Is the process of introducing biomolecules into cells. By using transfection reagents or specific plasmids, the cell will express the target molecule that will be packaged into EVs	Highly efficient large-load molecules	Could modify the EVs membrane structure	Variable	[107,108]
**Extracellular loading methods**					
**Sonication**	In order to permeabilize the EVs membrane and promote drug loading, it is used as an ultrasound probe with different amplitudes	Higher efficiency and continuous drug loading capacity	Causes EVs aggregation and affects the surface protein structure Induces membrane damage	Approximately 25%	[75,85,104,105,106,109]
**Electroporation**	Electrical field disturbs the phospholipid bilayer of vesicles, forming small pores in their membrane and thus allowing for the passage of the therapeutic agent into the EVs.	Simple to operate and ability to load large molecules (proteins]	Leads to RNA precipitation or EVs aggregation	Approximately 20%	[75,84,104,105,106,110]
**Freeze/Thaw cycles**	To allow drug entry, this process involves the formation of temporary pores on the EVs membrane through multiple rapid freeze–thaw cycles	Simple procedure and no change in EVs surface charge	Can induce EVs aggregation Encapsulation rate is generally lower	High drug delivery capacity	[75,104,105,106,111]
**Incubation**	Co-incubation of EVs with drugs at room temperature	The efficiency of packaging depends on the polarity of the therapeutic agent	The drug encapsulation rate is low	Approximately 15%	[75,104,105,106,112]
**Extrusion**	The drug is mixed with EVs and it is extruded with repeated steps; the EVs membrane deformation will allow for the entry of the drug.	The drug loading efficiency is high	Device-dependent process Disruption of the EV membrane	Approximately 23%	[75,104,105,106,113]
**Saponin**	Surfactant molecules that, when incubated with EVs, creates pores in their membranes through interaction with cholesterol.	Highly efficient	Difficult to remove completely Causes a continuous increase in EVs membrane permeability and cytotoxicity	Around 15%	[75,104,105,106,114]

**Table 2 biology-12-00665-t002:** Examples of the application of EVs in different clinical trials.

Study Title (NCT Number)	Status	Type of EVs	Cancer Model	Reference
Vaccination of metastatic melanoma patients with autologous dendritic cell (DC) derived exosomes	Completed	Autologous DEX	Metastatic melanoma	[129]
Dexosome immunotherapy in patients with advanced non-small cell lung cancer	Completed	Autologous DEX	Non-small-cell lung cancer	[130]
Trial of a Vaccination with Tumor Antigen loaded Dendritic Cell-derived Exosomes	Completed	Dendritic cell-derived exosomes loaded with antigen	Non-small-cell lung cancer	[95]
Immune responses in patients with esophageal cancer treated with SART1 peptide-pulsed dendritic cell vaccine	Completed	Autologous DEX	Esophageal cancer	[131]
Phase I Clinical Trial of Autologous Ascites-derived Exosomes Combined With GM-CSF for Colorectal Cancer	Completed	Ascites, autologous	Colorectal cancer	[128]
iExosomes in Treating Participants with Metastatic Pancreas Cancer with KrasG12D Mutation (NCT03608631)	Recruiting; phase I	Mesenchymal stromal cell- derived exosomes loaded with siRNA against KrasG12D	Metastatic pancreatic adenocarcinoma, pancreatic ductal adenocarcinoma	-
Pilot Immunotherapy Trial for Recurrent Malignant Gliomas (NCT01550523)	Completed	Tumor, autologous	Glioma	-
Edible Plant Exosome Ability to Prevent Oral Mucositis Associated with Chemoradiation Treatment of Head and Neck Cancer (NCT01668849)	Completed	Exosomes derived from plants	Head and neck cancer	-
Study Investigating the Ability of Plant Exosomes to Deliver Curcumin to Normal and Colon Cancer Tissue (NCT01294072)	Recruiting; phase I	Plant exosomes loaded with curcumin	Colorectal cancer	-

## Data Availability

Not applicable.

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
