# Peer review of "Extracellular Vesicles as Potential Therapeutic Messengers in Cancer Management"

_biology, 2023, doi:10.3390/biology12050665_

Round 1
Reviewer 1 Report
1.It is suggested to add new in vitro and in vivo research studies about the toxicological concerns.
2.what is the suggestion of this study for future works?
3.Please discuss and compare your results with previous works and add suggestions including Nano antioxidants based methods.
4.It will be better to add the role on mitochondria.
5.Please add details for time period and dose selection from literatures.
6.More references for the discussion part of manuscript and update and bold your study novelty should be added: e.g.,
-DOI: 10.1155/2021/1520052
-DOI: 10.1155/2021/4946711
-DOI: 10.1016/j.pestbp.2020.104586
Author Response
Please see the pdf file.
Thank you

Reviewer 2 Report
Major issues
The present review translates EVs as nanocarriers for treatment of (oncologic) diseases – but manages to only provide a general undetailed list of methods how this translation should or is implemented and some examples of effects of EV-treatments, without critical discussions or evaluations which are central to a review. In its current form, the manuscript provides low motivation for a reader to go through it.
Unclear definition of scope(s) from the beginning.
Unconnected excursion of historic anecdotes on EV discovery and extensive section about EV classification, questionable relevance to the current manuscript which deals with drug loading. In this context, EV types might be of interest if the loading is specific to EV type - the experience of the reviewer tells that drug loading is performed on bulk EVs usually.
Some methods are mentioned, but the most important aspects – performance comparison, yield, advantages/disadvantages, principle of action, current challenges etc. - are barely covered, just by some short notes in Table 1. From this few words, a reader does not get any useful or applicable information. Also the EV-drug loading methods listing is incomplete.
Clinical translation: partially the scope is missed – diagnostics rather than treatments are mentioned. The manuscript title clearly says “therapeutic messengers” and not “diagnostic ..”. More effort on clinical study evaluation needs to be performed to meet to scope of the manuscript.
Conclusions are made on issues that are nowhere mentioned in the manuscript.
Substantial proof reading by English native speaker required.
Specific comments:
The title should clearly indicate the scientific field the review deals with, in this case cancer and oncology obviously. In addition, the characteristics of EVs reviewed in the current manuscript in terms of pro-tumorigenic and niche-forming activity should be evident from the title to a potential reader.
Line 24: What is an “aggressive phenotype”? The reviewer guesses that oncogenic behaviour might me described by this, but the authors should make clear what they write about beforehand.
Line 57: communication requires also sending signals, not just receiving.
Line 211: Compounds like ..? The authors mentioned several times in the manuscript some factors, components or substances that there were investigated in studies, but forget to mention these factors, components or substances.
Line 215/216: As the review deals with nanomedicine, existing synthetic delivery systems should be extensively discussed and compared to EV-based delivery systems as advertised here.
Line 217-240: This section needs restructuring, as characteristics of EVs are seemingly repetitively described
Figure 1 should outline the described characteristics. Currently, it is a very basic figure that can be found in principle in any other work dealing with EVs.
Line 245: Critical information is missing here, as there was valid criticism whether the administered siRNA was truly EV-lumenal or just sticking to the outside.
Line 253: Selectively in comparison to which other cells or substances?
Section 2 should be separated into a section dealing with EV-loading methods and a section dealing with the effects of the cargo components in target cells subsequently. Section 2 as well as Section 2.1 is mixing up methods and description of therapeutic effects. What is missing is the critical evaluation of advantages and disadvantages of encapsulation methods and the reasons for that.
Line 290-293: As the scope of the review is oncology, this section is misplaced and was obviously only included as an easy to see through attempt to gain attention via the term “Covid-19”. Such attention-seeking is not scientific. Please remove and write another manuscript about Covid-19 if you really need to.
Line 347+: Although part of section 2.1 (intended to deal with encapsulation methods), this section deals with effects of autologous EVs rather than loading methods.
Table 1, Endocellular loading methods: This section is incomplete, there are many studies on donor cell-based RNA-loading into EVs, for example. Also proteins can be enriched via donor cell engineering. Please add this. Some specific notes:
“Electroporation - Ability to load large molecules” : How large? “Leads to RNA precipitation or
exosome aggregation” - How and why?
“Freeze / thaw cycles” - How are the physical properties of EVs changed? Studies report that EVs are destroyed by freeze-thawing.
“extrusion” - are there really still EVs present afterwards? Or is it just an undefined mixture of whatever? One can’t speak about “drug loading” when the EV membrane collapses and potentially never reforms.
“saponin” - which concentrations are used? Which methods of removing saponin after drug loading are there?
Line 386: “Specific” rather than “heterogeneous”. Never use the word eterogeneous in a diagnostic context, as it implies variability, which is detrimental for diagnostic applications.
Line 391: “and they can provide clinicians with a non-invasive biopsy” - this is already implied, no need to mention it again.
Line 395: “allow their detection at lower concentrations” - this does not depend on EV bilayer, the detectable concentration depends on the assay being used.
Line 396-405: This was already mentioned in a section above. Remove here or decide where to place this information.
Section 3: should the scope of this section include only translation of drug-loaded EVs in clinical studies on cancer? If yes, line 381-395 should be placed elsewhere or removed as the scope of the review encompasses according to the title “therapeutic messengers”, and not diagnostic aspects. Therefore, the scope of the review should be redefined to include diagnostics if intended, on the other hand diagnostics does not require drug loading into EVs and would probably be out of scope here.
Table 2: The reviewer is convinced that there are more than 3 clinical trials with EVs in cancer treatment are being currently conducted. Please invest more time and effort to reviewe the current landscape of clinical translation of drug-loaded EVs. In particular, the mentioned 3rd study of Table 2 involves an EV-loading strategy that was not covered in the methods listing of the current manuscript.
Conclusion:
Standardization was nowhere covered in the manuscript, so the authors can’t make a conclusion on it. The same applies to small/large scale production of EVs – production issues were nowhere discussed or even mentioned in the manuscript. “EV isolation” was not covered in the manuscript. “certain problems need to be solved before they can be used in clinical practice” - But there are already numerous clinical trials ongoin? “Thus, the scientific community should further focus on the exploration of these problems.” - What do the authors suggest in particular? Otherwise this is not a conclusion.
Author Response
Please see the pdf file.

Round 2
Reviewer 2 Report
The manuscript quality improved compared to version 1, however, there are still substantial issues to sort out before acceptance can be granted. Some sections are not relevant in the scope of the present review and the drawn conclusions are basically still not connected to the discussed content of the present review, while openly mentioning statements that were not discussed at all.
Please find the specific comments below:
Line 479: “Moreover, depending on the technique, EVs can be detected at lower concentrations, compared to traditional blood biopsies, presenting the capacity to modulate the phenotype of surrounding cells.” – This sentence is unclear. What is the connection of low EV concentration in relation to “capacity to modulate the phenotype” in a diagnostic context?
Line 469-483: If liquid biopsy is not in the scope of the present review, it should therefore not be included even as a side note. Rather than just mentioning 2 (randomly selected?) clinical trials in section 3, the authors should skip writing about liquid biopsy and write a well-structured informative section 3 on translation to the clinic instead. In particular, if the authors list several trials in Table 2! Why are they not discussed in the text? As a suggestion, the discussion of clinical trials could be categorized by the cancer type, EV source and/or loaded cargo in text and should involve a relevant number of trials. In the present form, section 3 is dispensable for a scientist to read and resembles more a short click bait news article.
Line 512: “The effect of tumoral cells derived-EVs varies according to the stage of tumor progression.” – This statement can not be a conclusion of the current review, as tumor progression was extensively discusse in the review.
Line 513: “production c” -> production
Line 516: omit “others”, because EVs are not synthetic delivery systems. Synthetic would be in vitro generate lipsomes eg.
Line 524-526: If not discussed in the review, no conclusion can be made. Omit this statements.
Line 527-533: Omit, as not relevent for the review.
Author Response
Please check the pdf file.
Thank you

Round 3
Reviewer 2 Report
The reviewer thanks the authors for adequately addressing the comments!